# Acute Toxicity of the Hydroethanolic Extract of the Flowers of *Acmella oleracea* L. in Zebrafish (*Danio rerio*): Behavioral and Histopathological Studies

**DOI:** 10.3390/ph12040173

**Published:** 2019-11-27

**Authors:** Gisele Custodio de Souza, Ianna Dias Ribeiro da Silva, Muller Duarte Viana, Nayara Costa de Melo, Brenda Lorena Sánchez-Ortiz, Monaliza Maia Rebelo de Oliveira, Wagner Luiz Ramos Barbosa, Irlon Maciel Ferreira, José Carlos Tavares Carvalho

**Affiliations:** 1Programa de Pós-Graduação em Inovação Farmacêutica, Departamento de Ciências Biológicas e da Saúde, Colegiado de Farmacia, Universidade Federal do Amapá, Macapá 68902-280, Brazil; custodio_gisele@yahoo.com.br; 2Laboratório de Pesquisa em Fármacos, Departamento de Ciências Biológicas e da Saúde, Colegiado de Farmácia, Universidade Federal do Amapá, Macapá 68902-280, Brazil; mvianna987@gmail.com (M.D.V.); nayara.unifap@yahoo.com.br (N.C.d.M.); charmed1797@gmail.com (B.L.S.-O.); 3Programa de Pós-Graduação em Ciências Farmacêuticas, Faculdade de Farmácia, Universidade Federal do Pará, Belém 66075-110, Brazil; iannadrs@gmail.com (I.D.R.d.S.); monalizamaia@yahoo.com.br (M.M.R.d.O.); zweigw@gmail.com (W.L.R.B.); 4Laboratório de Biocatálise e Síntese Orgânica Aplicada, Departamento de Ciências Biológicas e da Saúde, Colegiado de Farmácia, Universidade Federal do Amapá, Macapá 68902-280, Brazil; irlon.ferreira@gmail.com

**Keywords:** *Acmella oleracea*, *Danio rerio*, oral and immersion, toxicity, histopathology

## Abstract

Hydroethanolic preparations of the botanical species *Acmella oleracea* L. are used in the north of Brazil for the treatment of various diseases. However, few studies have been conducted to evaluate the toxicity of this species. The objective of this study was to evaluate the acute toxicity of the hydroethanolic extract of *A. oleracea* L. (EHFAo) flowers in zebrafish by immersion and oral administration. The extract was analyzed by ultra-performance liquid chromatography–mass spectrometry (UPLC–MS). EHFAo was administered orally (44.457, 88.915, 199.94, 281.83, and 448.81 mg/kg) and by immersion (250, 300, 350, 400, and 450 µg/L). Behavioral and histopathological analysis of gills, liver, intestine, and kidney were performed. The presence of (2E,6Z,8E)-N-isobutyl-2,6,8-decatrienamide (spilanthol) in EHFAo was identified by ultra-high-re.solution liquid chromatography–electrospray ionization mass spectrometry (UHPLC–ESI-MS). Treatment with EHFAo caused significant behavioral changes and death. The calculated median lethal dose (LD_50_) was 148.42 mg/kg, and the calculated median lethal concentration (LC_50_) was 320 μg/L. In the histopathological study, it was observed that upon oral treatment, the tissue alterations that compromised the normal functioning of the organism occurred with EHFAo doses of 88.915, 199.53, and 281.83 mg/kg, the intestine being the most affected. When the treatment was performed by immersion, the most toxic EHFAo concentrations according to the histopathological evaluation were 300, 350, and 400 μg/L, with the most affected organ being the gills. Finally, EHFAo in this study was shown to be more toxic to the liver, intestine, and kidneys when administered orally and to gills, liver, and kidneys when administered by immersion in water. Therefore, considering the results obtained and the chemical characteristics of the main phytochemical marker of EHFAo, spilanthol, it can be suggested that, depending on the dose, this compound can lead to histopathological damages in the organs highlighted in this study.

## 1. Introduction 

Throughout the history of humankind, plants have been the basis of medical treatments. They are used to produce herb infusions, ointments, and balsams, and their activities are related to their active compounds according to scientific knowledge. Already more than 3000 active substances used in traditional medicine have been registered [1]. Several of these substances are derived from plant extracts that, after being metabolized, are excreted in the feces or urine, entering the aquatic environment and still carrying active molecules [2].

*Acmella oleracea* (L) R. K. Jansen is a plant species popularly known as jambu. This species belongs to the family Asteraceae, is native to the Eastern Amazon, being cultivated on a large scale in the Brazilian states of Pará and Amapá, and is widely used in folk medicine [3,4]. The leaves and inflorescences have been used to treat diseases of the mouth and throat [5,6], as a diuretic [4,7,8], for influenza and cough, as an antibacterial, antifungal, antimalarial [9,10,11], for the treatment of rheumatisms [12], as an insecticide [9,13], and as an anti-inflammatory, analgesic, and local anesthetic [14,15,16,17,18,19]. The inflorescence is the part of the plant most used as a local anesthetic for toothaches [3].

The main chemical constituents found in *A. oleracea* flowers are alkylamides, particularly spilanthol (2E,6Z,8E)-N-isobutyl-2,6,8-decatrienamide, known for its pharmacological properties [20,21,22].

The search for new therapies that may help in the treatment of various diseases has grown, and zebrafish is used as a model organism for the evaluation of the toxicity of substances of natural and synthetic origin [23].

Among the several methodologies applied when using zebrafish, acute toxicological studies by immersion are widely employed for environmental evaluations and have demonstrated efficacy in the evaluation of the toxicological potential of substances diluted in water based on histological analyses of the gills [24,25]. Oral administration by gavage is an innovative methodology to evaluate the toxicity of several substances with pharmacological potential in zebrafish, having as the first contact organ the intestine [26,27,28]. The objective of this study was to evaluate the acute toxicity of the hydroethanolic extract of the flowers of *A. oleracea* L. (EHFAo) in zebrafish by both immersion, using different concentrations diluted in the water of the maintenance system, and oral administration, in an attempt to elucidate the toxicological potential of this extract.

## 2. Results 

### 2.1. Analyses by Ultra-High-Resolution Liquid Chromatography–Electrospray Ionization Mass Spectrometry (UHPLC–ESI-MS)

EHFAo analyzed by UHPLC–ESI-MS in positive ionization mode showed a peak at 2.64 min in the total ion chromatogram (TIC), indicating an abundance of 22.56% (peak integration) for the molecule (2E,6Z,8E)-N-isobutyl-2,6,8-decatrienamide (spilanthol), as well as the presence of other compounds in the sample (Appendix A) [29].

### 2.2. Behavioral Analysis

Treatment of zebrafish at the oral doses of 44.457, 88.915, 199.94, 281.83, and 448.81 mg/kg and at the water concentrations of 250, 300, 350, 400, and 450 μg/L) triggered significant behavioral changes in the animals, as shown in Figure 1A,B. The percentage of alterations was higher for the three higher doses (199.53, 281.83, and 448.81 mg/kg) and concentrations (350, 400, and 450 μg/L) (Table 1 and Table 2). Signs of stress were recorded as spasms, tail tremors, loss of posture and motility, permanence at the bottom of the aquarium, and death.

### 2.3. Determination of LD_50_ and LC_50_

Figure 2A shows that the mortality of the animals was dose-dependent, with the lowest dose (44.447 mg/kg) causing 8.3% of deaths, and the highest dose (448.81 mg/kg) causing 100% of deaths. The calculated median lethal dose (LD_50_) was 148.42 mg/kg.

In the immersion-treated animals, it was observed that the mortality pattern was similar to that of the orally treated animals (Figure 2B). As the concentration of the extract increased in the water, also the mortality increased. The concentration of 250 μg/L caused mortality only in 16.6% of the animals, while the highest concentration (450 μg/L) killed 100% of the animals. The calculated median lethal concentration LC_50_ was 320 μg/L.

### 2.4. Histopathology

The oral and immersion treatments of zebrafish with EHFAo produced histopathological changes in the gills, liver, intestine, and kidneys. From the systematization of these changes, it was possible to calculate the index of histopathological changes (IHC), represented in Figure 3, Figure 4, Figure 5 and Figure 6.

As can be seen in Figure 3 and Figure 5 and Table 3 and Table 4, treatments with EHFAo caused histopathological changes in the gills, as indicated by the calculated indices (Figure 4 and Figure 6). However, the animals treated orally presented only histopathological alterations of stage I, whose IHC characterizes the organ as functionally normal. The same was observed in the gills of the animals immersion-treated with EHFAo concentrations of 250 μg/L and 450 μg/L. The presence of mucosal cells, hypertrophy and/or hyperplasia of chloride cells, and hyperplasia of epithelial cells in the base of secondary lamellae was observed (Figure 3 and Figure 5). No alterations were observed in the gills of control animals.

Concentrations of 300, 350, and 400 μg/L of EHFAo produced moderate to severe changes in the gills of these animals, according to the IHC presented in Figure 6. We found a series of histopathological changes in the gills (Table 4) characterized by all three stages of change. Hyperplasia at the base of the secondary lamellae, lamellar epithelium displacement, chloride cell hyperplasia, and blood vessel dilatation were the most frequent stage I changes. The complete fusion of some secondary lamellae (Figure 5B), epithelial rupture, and cell degeneration were the most frequent stage II alterations. Necrosis was the only stage III alteration.

When the IHC of the gills after the administration of the oral doses of EHFAo was compared with that of the control group (saline solution) and between doses (Figure 4), the results were not statistically significant (*p* > 0.05). In the treatments by immersion, only the IHC corresponding to the EHFAo concentration of 450 μg/L showed no significance when compared to that of the control (Figure 6). For the other concentrations of EHFAo (250, 300, 250, and 400 μg/L), the IHC presented statistically significant values (*p* < 0.05) when comparing to the control and between concentrations.

In the liver, several histopathological changes (Table 3 and Table 4) occurred upon both oral and immersion treatments, as shown by the measured IHC in Figure 4 and Figure 6. Figure 4 shows that only animals treated with the highest oral dose (448.81 mg/kg) underwent mild to moderate liver changes, as the other doses (44.457, 88.915, 199.53, 281.83 mg/kg) were associated with IHC indicating moderate to severe liver alterations.

Decreases in the relative frequency of nuclei, cytoplasmic vacuolization, and glycogen were the most frequent stage I alterations in all groups. The most frequent stage II alterations were nuclear degeneration and cellular degeneration. Focal necrosis was the only stage III alteration recorded (Figure 3). Figure 3B shows a healthy zebrafish liver.

In the immersion teratment, only the liver of the animals treated with EHFAo concentrations of 250 and 450 μg/L led to IHC (Figure 6) indicating mild to moderate organ alterations. The other concentrations (300, 350, and 400 μg/L), were associated with IHC values indicating moderate to severe organ changes, as can be seen in Figure 6.

The most frequent stage I alterations in immersion-treated animals were hepatic cord disorganization, nuclear atrophy, intense cytoplasmic vacuolization, and increased cell volume (Figure 6). The most frequent stage II alterations were cell degeneration and hyperemia. Necrosis was also recorded as a stage III alteration.

The IHC of all groups of animals treated orally and by immersion were statistically significant when compared to that of the control, with *p* < 0.05 (Figure 4 and Figure 6).

The intestine of *Danio rerio* has villi composed of epithelial cells and goblet cells, protected by an important mucosal barrier (Figure 3A). In this study, all treatments with EHFAo caused changes in the intestinal tissue of the animals, as can be seen in Figure 3 and Figure 5 and Table 3 and Table 4.

Figure 4 shows that the intestinal IHCs of all groups (44.457, 88.915, 199.94, 281.83, and 448.81 mg/kg) of orally treated animals indicated moderate to severe organ alterations. In contrast, uon the immersion treatments, all groups (250, 300, 350, 400, and 450 μg/L) presented IHC which classified the organ as normal.

For orally treated animals, muscle layer degeneration, goblet cell hyperplasia, leukocyte infiltration, and edema were the major stage I changes recorded. The most frequent stage II alterations were: cellular degeneration, villous atrophy, and vacuolization of enterocytes. Stage III alteration was necrosis (Figure 3). However, in the immersion-treated animals, only stage I alterations, such as hyperplasia of the epithelial cells and detachment of the epithelial sheet, were recorded, as well as partial fusion of the villi (Figure 5).

All the IHC values of the orally treated groups were statistically significant when compared to the control and among them, with *p* < 0.05. When considering the immersion treatement, the IHC values (Figure 4) of none of the treated groups was significant with respect to the control and when comparing between the groups, with *p* > 0.05.

In Table 3 and Table 4 and Figure 3 and Figure 5, it is possible to observe the most frequent alterations found in zebrafish kidneys treated with EHFAo. In the present study, alterations caused by the oral treatment were more severe for the kidneys than for the other organs, as it was observed that the doses of 88.915, 199.52, and 281.83 mg/kg led to high IHC (40.99, 38.41, and 41 respectively), indicating moderate to severe organ alterations. At doses of 44.473 and 448.81 mg/kg, the IHC values indicated mild to moderate organ changes.

The severity of damage in the zebrafish kidneys represented by the IHC was also observed in the immersion groups, where all groups (250, 300, 350, 400, and 450 μg/L) presented indices that indicated moderate to severe organ alterations (Figure 6).

Thus, various alterations were recorded for this organ, and the main stage I changes observed in the groups treated with EHFAo by both routes of administration were: mild tubular hyaline degeneration, tubular cell hypertrophy, and tubular lumen increase. Cytoplasmic degeneration of tubular cells, severe hyaline tubular degeneration, degeneration of the tubules and glomeruli were the most evident Stage II changes. Necrosis was the most severe stage III alteration and was also recorded (Figure 3 and Figure 5).

According to the IHCs (Figure 4 and Figure 6) calculated, it was observed that treatment with EHFAo at all doses and concentrations tested caused alterations that were statistically significant when the doses and concentrations were compared to the control and among them *p* < 0.05.

## 3. Discussion

In order to evaluate the acute toxicity of a hydroethanolic extract of the flowers of *A. oleracea* (L.) in zebrafish by immersion and oral administration, the toxicological potential was clarified at the histopathology level. Thus, it was observed that the different treatment routes had toxic effects for specific organs.

By analysis with UHPLC–ESI-MS, the presence of (2E,6Z,8E)-N-isobutyl-2,6,8-decatrienamide (spilanthol) was identified in the EHFAo [29]. This alkylamide is part of a group of compounds consisting of the union of a medium-to-large fatty acid, with 8 to 18 carbons, generally aliphatic, and one amine [30]. The molecular characteristics of this compound, which is the major EHFAo component, favor easy absorption through the skin and the intestinal mucosa [29].

The results of this analysis showed that the pharmacological activity of *A. oleracea* species is due to spilanthol, which, according to Dubey et al. [31], has many pharmacological properties, of which the anesthetic activity is the most prominent [32,33,34,35]: it also have antioxidant [36], anti-inflammatory [37], anti-wrinkle [38], antifungals [31], antimalarials [35] activities. There are also reports indicating anticonvulsant, antioxidant, aphrodisiac, pancreatic lipase inhibitor, antimicrobial, antinociceptive, diuretic, vasorelaxant [31] properties.

After administration of EHFAo either orally or by immersion, the animals presented behavioral changes in all treatment groups. These results are similar to those of Ribeiro [39], who stated that the hydroethanolic extract of the jambu (*Spilanthes acmella*) roots in zebrafish can alter the behavior, and these behavioral changes begin with the increase of the swimming activity, which, according to Little et al. [40], is an indicator of the overall internal status of the animal. Exposure of the animal to a stress situation triggers its first defense mechanism, which most often is an escape behavior, to reduce the likelihood of death [25]. This behavior was also observed in the orally treated animals.

Everds et al. [41] state that in animal toxicity tests, animal stress is common and may lead to changes in body weight, food consumption, behavior, blood circulation, and reproductive functions. However, not all of these factors are usually evaluated in specific studies.

Mortality was evident in all animals treatmed with EHFAo. In a study by Santos et al. [23], it has been observed that, as the concentration of potentially toxic substances in the water increases, the mortality of animals increases. The calculated oral LD_50_ was 148.424 mg/kg, which is lower than that calculated for rats in a study by Nomura et al. [42].

Internally, other damages may occur, both in dead animals and in survivors, such as histopathological changes in different organs [24,25]. These damages were also caused by the oral treatment. It is known that zebrafish has, on each side of the pharynx, four branchial arches, each with two rows of filaments, which have on each side the secondary lamellae. It is a bilateral organ located outside the opercular cavity [43].

In the animals treated by immersion with EHFAo, it was observed that mortality was more evident at the highest concentrations (400 and 450 μg/L) and occurred in the first hour of treatment. Souza et al. [24] and Barron end Hoffman et al. [44] explained that gills are considered as an organ of dominance in the removal of substances from water because they have a large surface of absorption. They are particularly sensitive to toxic substances because of their direct contact with water during gas exchange [45]. In the orally treated animals, the IHC for the gills was not statistically significant.

The most frequent change in all treatment groups with EHFAo was the displacement of epithelial cells. Several authors affirm that the displacement or elevation of epithelial cells is one of the first histopathological changes observed in the gills of fish that have been exposed to toxic agents [24,25,46,47,48]. According to Borges et al. [26], this change indicates an attempt of aquatic animals to adapt to new pathophysiological conditions. The space formed between the lamella and the displaced epithelium can fill with water, which leads to the formation of edema. These changes may lead to dysfunction of the gills and suffocation [49,50,51].

Lamellar hyperplasia, fusion, and chloride cell hyperplasia have been widely observed in animals treated by immersion with EHFAo and are considered, according to Rigolim-Sá [46], as mechanisms of defense of the gills, which promote an increase of the blood–water barrier; it is considered an initial response of the gill apparatus, characterized by increased tissue cellular functions caused by changes in physiological activities. Although it is a defense mechanism, reducing or even totally hampering the passage of water between the secondary lamellae, this loss of respiratory surface can cause death by anoxia [28,52,53,54,55].

Epithelial rupture and cellular degeneration are considered regressive alterations, caused by tissue hypofunction [44]. These same authors reported that many pathological agents might cause changes in the gill tissue, such as vacuolization and necrosis of secondary lamellae, which were observed in this study. Only the groups treated with EHFAo concentrations of 300, 350, and 400 μg/L showed cellular degeneration and necrosis. This alteration can be caused by loss of function in the gill tissue [56] and exposure to conditions of higher toxicity [57].

High concentrations of a toxic substance may cause high IHC. However, this is only possible if the test substance is sufficiently toxic without causing mortality in the first few hours of treatment [24,25,26]. This relationship was not observed for the highest concentration of EHFAo administered by immersion, since it was lethal in the first hour of treatment. Santos et al. [25] reported that the time of occurrence of the lethal effect of a test substance might influence the occurrence of tissue damage to any organ. By the oral route, the IHC in the gills were not significant compared to that of the control, as the gills were not the first organ in contact with EHFAo.

The liver of the zebrafish is similar to that of mammals in the main physiological processes performed, although its structure is different. These include drug metabolism pathways, comprising the action of cytochrome P450 that allows metabolic reactions, such as hydroxylation, conjugation, oxidation, demethylation, and unetilation. Also, the liver is the most important site for biosynthesis and biotransformation, being essential for bile synthesis, storage of lipids and glycogen, as well as for the production of vitellogenin, a protein present in the film that surrounds the egg [58]. Therefore, after exposure to toxic substances, its histopathology can be compared to that of mammals [26,28,59]. The histopathological changes caused by EHFAo can alter liver’s normal functioning, resulting in a lower metabolic potential as well as in a decrease in its glycogen storage capacity.

Vacuolation, cellular degeneration, and hyperemia were observed in animals treated with the highest EHFAo doses (199.53, 281.83, and 448.81 mg/kg) and concentrations (350, 400, and 450 μg/L). In a study in zebrafish with oral nanoemulsions, Borges et al. [26] stated that liver alteration may be related to the reduction of glycogen stores in hepatocytes and to the accumulation of lipids combined with toxic agents, which may alter the normal functioning of the organ. Hyperemia occurs as an attempt to increase the general blood flow in the liver and increase the release of nutrients and oxygen to the affected areas, avoiding hypoxia [24,49,56]. Hepatic necrosis has also been reported by Borges et al. and was caused by high doses of the toxic agents [26].

In this study, the intestine of zebrafish was significantly affected in the orally treated groups, since it was the first organ coming in close contact with EHFAo. According to Carvalho et al. [49], zebrafish intestine is formed by a mucosal layer with goblet cells, inflammatory cells, and enterocytes with functions that go beyond the absorption of nutrients; the intestinal epithelium is a site of immune responses and control of osmotic balance [26] and a recycling site for enzymes and macronutrients [60].

Until recently, studies involving the histology of zebrafish gut were based only on observations [61]. However, recent studies have broadened the knowledge about the histopathological changes that can occur in the intestine of this animal [26,49]. Exposure to toxic substances causes damage to the intestinal mucosa and to cellular development, which may disturb the physiology of the organ and cause various histopathological changes [49,56].

In all groups of animals treated with EHFAo by the oral route, displacement of the intestinal mucosa, leukocyte infiltration, and lymphocytic infiltration were observed in the mucosal layer. Borges et al. [26] also observed these alterations in animals orally treated with nanoemulsions based on *Rosmarinus officinalis* L. This result indicates that EHFAo was toxic because it caused an increase of the number of defense cells in the intestinal epithelium, which may be related to the development of inflammation in the lamina propria [49,61]. Vacuolization was also observed in the treated groups. This change is common after exposure to substances with high toxicity and usually precedes necrosis [56]. This fact justifies the presence of intestinal necrosis in all groups orally treated with EHFAo.

The kidney of the adult *Danio rerio* contains nephrons that are responsible for the filtration of blood residues and the absorption of salt and water. It presents renal corpuscles and proximal and distal convoluted tubules (Figure 3A). The Zebrafish’s kidney plays the important function of excreting water entering the fish through the mouth. It also performs the filtration of residues and the absorption of salt and water [26,55,56]. It is one of the organs most affected by toxic substances according to Carvalho et al. [49]. In this study, it was the organ most affected by both immersion and oral treatments.

The groups of animals treated at all doses and concentrations of EHFAo showed hypertrophy of tubular cells in the renal tissue. According to Carvalho et al. [49], this condition occurs as a consequence of the dryness of renal tubule epithelial cells, which in some cases, as in this study and in a study performed by Borges et al. [26], may precede hyaline degeneration, which consists of an increase of the amount of eosinophilic granules in the cytoplasm of these cells [49]. Hyaline degeneration, a condition observed in this study, may be related to the reabsorption of excess proteins synthesized by the glomerulus [56].

Hyperemia, tubular disorganization, tubular degeneration, and cytoplasmic degeneration of tubular cells were observed in all groups treated with EHFAo at different doses and concentrations. Hyperemia is an increased amount of circulating blood and may be associated with vessel rupture [26]. In the kidneys, it can be caused by the pressure exerted by the dilation of glomerular capillaries in the presence of toxic substances [49]. According to Carvalho et al. [49], tubular changes observed in zebrafish kidneys may be indirectly caused by the metabolic dysfunction induced by exposure to toxic substances. These changes can often culminate in kidney necrosis [56]. This explains the presence of necrosis in zebrafish kidneys treated with EHFAo orally and by immersion.

## 4. Material and Methods

### 4.1. Plant Material

Samples of the flowers of *A. oleracea* (L.) RK Jansen were collected in September 2016 in Fazendinha district (S 0 ° 02’30.40 ‘’/W 5106’37.5 ‘’), in the city of Macapá, State of Amapá, Brazil. An exsiccata of the species was deposited in the IAN Herbarium (Embrapa Amazônia Oriental, Bélem, Pará, Brazil) under identification number 196011.

### 4.2. Obtaining the Hydroethanolic Extract of the Flowers of A. Oleracea (L.) R. K. Jansen (EHFAo)

The Fresh flowers, previously selected, were milled to obtain a fine granulation powder, then the material was cold-macerated for seven days in a 70% hydroethanolic solution. The resulting extractive solution and the macerate were filtered and then concentrated on a rotary evaporator (Quimis Model Q 218.2) at 40 °C for complete evaporation of the solvent. Subsequently, the concentrate was lyophilized, yielding a 2.5% yield [29]. EHFAo analysis was performed by UHPLC–ESI-MS, as described by Souza et al. [29].

Samples containing 5 mg/mL of the extract were prepared with methanol, filtered in microfilters, and then analyzed on a reverse-phase column (ZORBAX XDB C8; 2.1 × 50 mm 3.5micron), eluted with water and (A) 0.1% acetic acid and (B) acetonitrile (40:60) in isocratic mode, with 2 μL of injection volume, flow rate of 0.05 mL/min, and 1.200 bars of pressure limit, in 13 min. The column temperature was kept at 40 °C, the thermostat at 20 °C, and the samples were kept at room temperature. The compounds were detected at 230 nm. Mass spectrometry was performed through electrospray ionization in full scan mode, operating between 50 and 700 *m*/*z*, with 50 V of collision energy. Nitrogen gas was used as nebulizer (45 psi), with a flow rate of 5 L/min in positive mode. The mass found was registered in positive ionization mode, and the spectra of the fragments were identified according to the literature.

### 4.3. Acute Toxicity Study of EHFAo

#### 4.3.1. Animals

Zebrafish (*D. rerio*) of the AB wild line (± seven months) were obtained from Acqua New Aquarium and Fish Ltd.a. ME, Igarassu-PE, Brazil, and after a quarantine period, were packed in the Zebrafish Platform of the Federal University of Amapá (UNIFAP), in a recirculation system under a photoperiod of 14/10 h day/night. Water parameters, including temperature (26 ± 2 °C), pH (6.0–8.0), conductivity (8.2 ± 0.2), and cleaning of the recirculation system were monitored daily. The animals were fed twice a day with *Artemia salina* in the morning and with a commercial ration (Tetramim) in the afternoon. The initial project was approved by the Ethics Committee of Animals Use from the Federal University of Amapá, CEUA/UNIFAP (2018/002).

#### 4.3.2. Experimental Design

The animals used in this study were orally treated with the doses of 44.457 mg/kg, 88.914 mg/kg, 199.54 mg/kg, 218.83 mg/kg, and 448.81 mg/kg (diluted in 1 mL of saline solution) and by immersion at concentrations of 250 µg/L, 250 µg/L, 300 µg/L, 350 µg/L, 400 µg/L, and 450 µg/L (diluted in the maintenance system water); for each treatment route, 5 groups with 4 animals were used in 3 replicates (*n* = 12 animals/group), weighing between 300 and 400 mg. 

#### 4.3.3. Behavioral Analysis and Mortality

Behavioral reactions were classified in three stages: (1) increased swimming activity, spasms, and tremors in the tail axis; (2) circular movement and loss of posture; (3) clonus, loss of motility, deposition of the animal on the bottom of the aquarium, and death. Each animal was evaluated individually and was considered dead when the movement of the operculum and the response to mechanical stimulation could no longer be detected [24]. The other animals were submitted to euthanasia through anesthetic cooling, following the recommendation of the American Guidelines of the Veterinary Medical Association for Animal Euthanasia [62].

#### 4.3.4. Determination of LD_50_ and LC_50_

Adult animals collected at random from the maintenance system were fasted for 24 h before oral treatment with EHFAo at the doses of 44.457, 88.914, 199.53, 281.83, and 448.81 mg/kg. The animals were weighed and immobilized in a sponge, and, with the aid of a volumetric pipette (HTL Lab Solutions), the oral treatment was administered, at a maximum volume of 1.5 μL/animal [63,64].

For the LC_50_ determination, the concentrations (250, 300, 350, 400, and 450 μg/L) of EHFAo were diluted in 1 L of water from the maintenance system, where they remained for 48 h.

After treatment, the animals were observed for 48 h for behavioral evaluation and mortality. The LD_50_ and LC_50_ were determined using the probit method [26].

#### 4.3.5. Histopathological Analysis

For histopathological analysis of the organs (branchial, liver, intestine, and kidneys), the animals were fixed in Bouin solution for 24 h and then decalcified in EDTA (ethylenediamine tetraacetic acid, Sigma Co., São Paulo, Brazil) for 24 h. The samples were successively dehydrated in a graded alcohol series of 70, 80, 90, and 100%. They were diaphonized by impregnation with xylol and embedded in paraffin. The samples were sectioned at 5 μm using a microtome (Brand Rotary Microtome Cut 6062, Slee Medical, Berlin, Germany), and histological analysis was performed after the tissue sections were stained with hematoxylin and eosin as described by Souza et al. [24], Carvalho et al. [49], and Borges et al. [26]. The images were analyzed using an Olympus Microscope BX41-Micronal and photographed with an MDCE-5C USB 2.0 (digital) camera.

#### 4.3.6. Assessment of Histopathological Changes

The index of histopathological changes (IHC) was calculated from the stages of tissue changes observed in the gills, liver, kidneys, and intestine (Appendix A). Alterations can be classified as stage I, II, and III, and the IHC value indicates whether an organ is healthy (0 to 10), moderately altered (11 to 20), with moderate to severe changes (21 and 50), or containing severe irreversible changes (>100) [43,46,49,65]. Thus, the indices were calculated according to the following equation:(1)$$I\;=\;\frac{\overset{na}{\underset{i-1}{\sum }}ai\;10\;\overset{nb}{\underset{i-1}{\sum }}bi\;+\;10^{2}\overset{nc}{\underset{i-1}{\sum }}ci}{N}$$
where a: first-stage changes; b: second-stage changes; c: third-stage changes; na: number of changes considered as first-stage changes; nb: number of changes considered as second-stage changes; nc: number of changes considered as third-stage changes; N: number of fishes analyzed per treatment.

#### 4.3.7. Statistical Analysis

The median lethal dose (LD_50_) and the median lethal concentration (LC_50_) were determined by probit analysis. The statistical analysis was performed using the software GraphPad Prism (v 6.0). The comparison between groups was performed using one-way ANOVA followed by the post-hoc Tukey–Kramer test. The results are presented as mean ± standard deviation of the mean (SD), and values with *p* < 0.05, *p* < 0.01, and *p* < 0.001 were considered statistically significant.

## 5. Conclusions 

The treatment of *D. rerio* with EHFAo orally and by immersion for 48 h induced behavioral changes in zebrafish. The most altered organs in the histopathological study upon oral treatment were liver, intestine, and kidneys; the gills had the smallest IHC. In immersion-treated animals, the most affected organs were gills, liver, and kidneys. Therefore, considering the results obtained and the chemical characteristics of the main phytochemical marker of EHFAo, spilanthol, it can be suggested that, depending on the dose, this compound can cause histopathological damages in various organs, as reported in this study.

## Figures and Tables

**Figure 1 pharmaceuticals-12-00173-f001:**
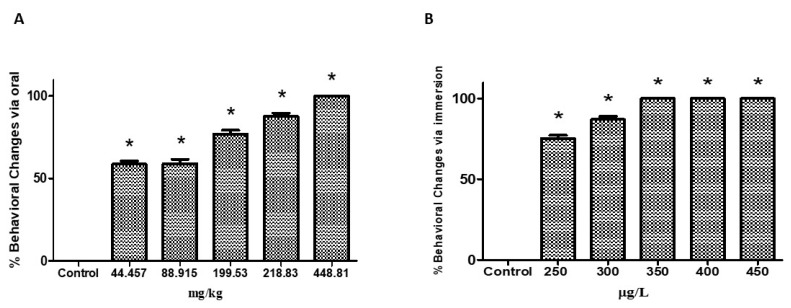
Percentage of behavioral changes in animals treated with different oral doses and water concentrations of the hydroethanolic extract of *Acmella oleracea* L (EHFAo). The behavioral alterations were significantly increased in the animals treated with the extract either orally (**A**) or by immersion (**B**) compared to the control. Data are presented as mean ± SD. Statistical analysis was performed through one-way ANOVA followed by post-hoc Tukey test; * *p* < 0.001 denotes significance to control.

**Figure 2 pharmaceuticals-12-00173-f002:**
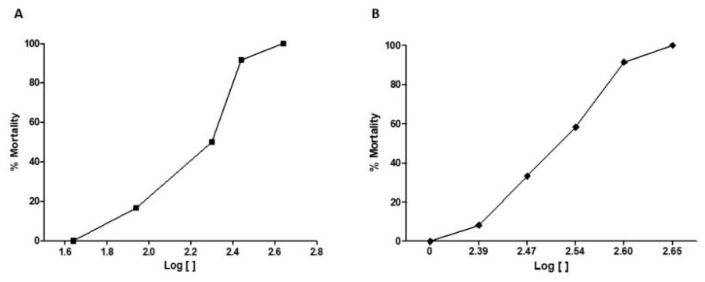
Effect of the treatment with different (**A**) oral doses (44.457, 88.914, 199.53, 281.83, and 448.81 mg/kg) and (**B**) water concentrations (250, 300, 350, 400, and 450 µg/L) of EHFAo in zebrafish, *n* = 12 animals/group. Median lethal dose (LD_50_) = 148.424 mg/kg. Median lethal concentration (LC_50_) = 320 µg/L.

**Figure 3 pharmaceuticals-12-00173-f003:**
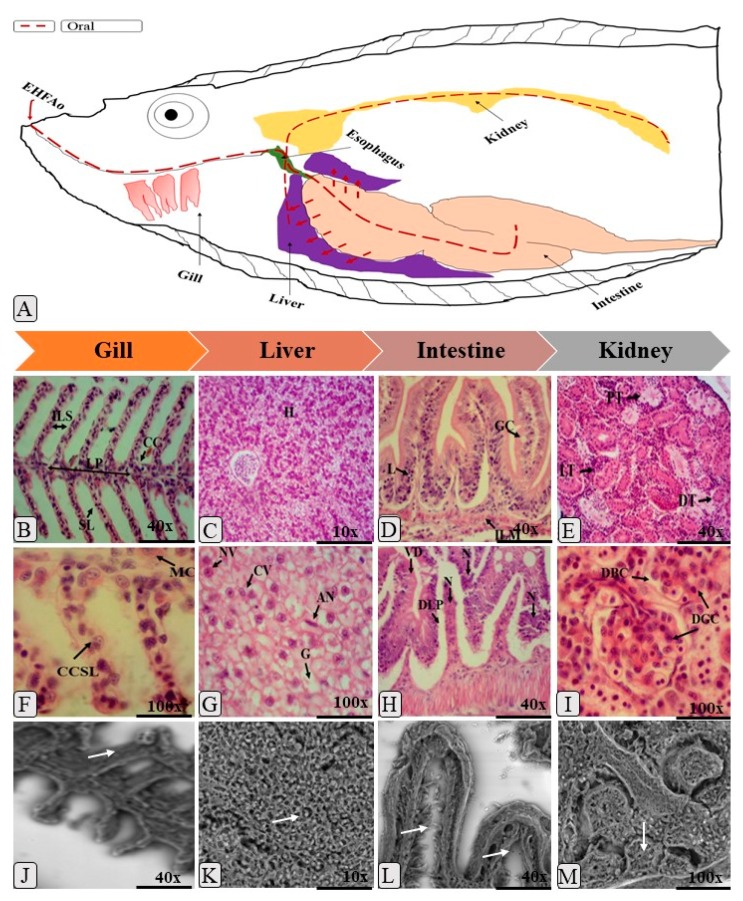
(**A**), schematic drawing of the metabolism of orally administered EHFAo in zebrafish; (**B**), hematoxylin and eosin (H&E) staining of a longitudinal section with normal branchial filaments; ILS: space between secondary lamellae, CC: chloride cells, SL: secondary lamella, and LP: primary lamella; (**F**), (H&E staining), mucosal cell (MC) and chloride (CCSL); (**J**) (SEM: scanning electron microscopy) dilatation of lamellar epithelium (white arrow); (**C**), (H&E staining), longitudinal section of normal liver; (**H**) hepatocytes; (**G**) (H&E staining) and (**K**) (SEM) cytoplasmic vacuolization (CV and white arrow) is observed; (**D**), (H&E staining) detachment of lamina propria (DLP) and villi degeneration (LV), in (**L**) (SEM) and observed in the intestine of zebrafish; GC: goblet cells, ILM: muscle layer, L: lymphocytes; white arrow: villous atrophy; a normal longitudinal section of the zebrafish kidney is shown in (**E**) (H&E staining), where it is observed in PT: proximal tubule, DT: distal tubule, and LT: lymphoid tissue; (**I**), (H&E staining) dilation of glomerular capillaries (DGC) and decreased Bowman’s capsule space (DBC) are observed, (**M**) (SEM), hyaline degeneration (white arrows) is observed.

**Figure 4 pharmaceuticals-12-00173-f004:**
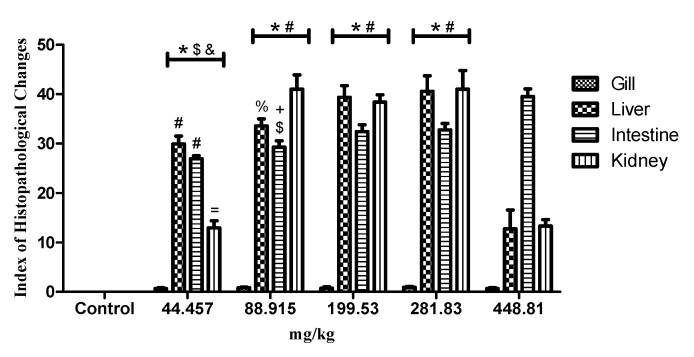
Index of histopathological changes for each EHFAo dose. Data are presented as mean ± SD (*n* = 4/groups); * *p* < 0.001 vs. control, # *p* < 0.001 vs. 448.81 mg/kg, $ *p* < 0.001 vs. 281.83 mg/kg, & *p* < 0.001 vs. 199.53 mg/kg; % *p* < 0.05 vs. 281.83 mg/kg, + *p* < 0.05 vs. 199.53 mg/kg, = *p* < 0.05 vs. 88.915 mg/kg. Statistical analysis was performed through one-way ANOVA followed by the posthoc Tukey test.

**Figure 5 pharmaceuticals-12-00173-f005:**
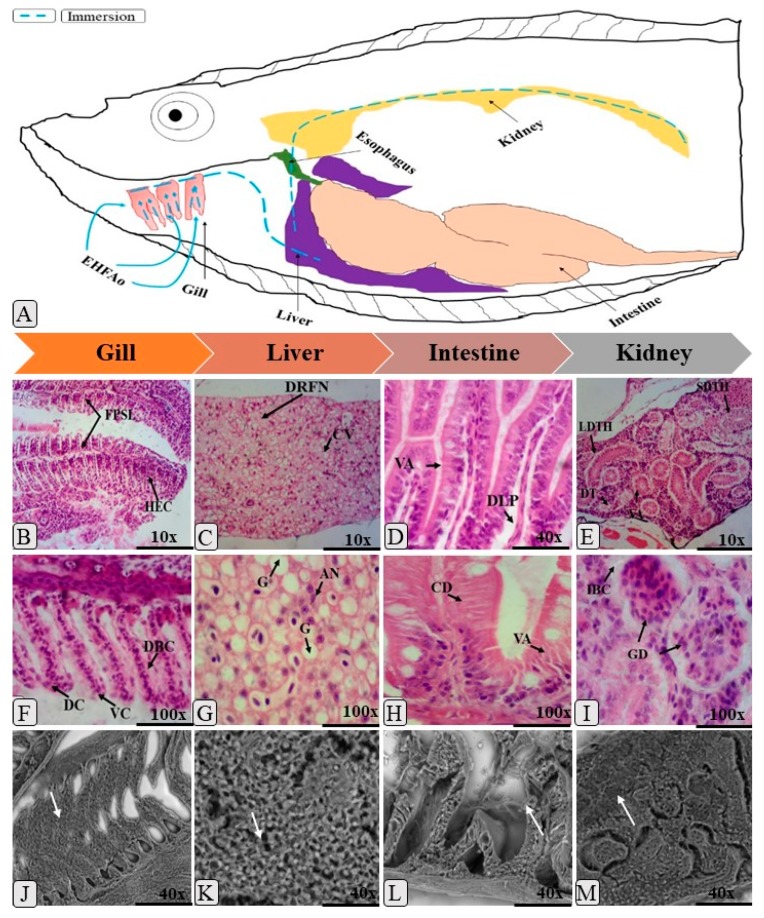
(**A**) Schematic drawing of the metabolism of EHFAo in zebrafish upon treatment by immersion. The complete melting of the secondary lamellae (FPSL and white arrow) and hyperplasia epithelial cells (HEC), capillary dilatation (CV), and vascular congestion (CV) are observed in (**B**), (**F**) (H&E staining), and (**J**) (SEM); in the case of (**C**), (**G**) (H&E staining), and (**K**) (SEM), we observed a longitudinal section of the liver with decrease in the relative frequency of nuclei (DRFN), nuclear atrophy (AN), and glycogen (G); (**D**), (**H**) (H&E staining), and (**L**) (SEM), a longitudinal section of the intestine where atrophy of the villi (VA and white arrow), detachment of lamina propria (DLP), and cell degeneration (CD) are observed; (**E**), (**I**) (H&E staining), and (**M**) (SEM), a longitudinal section of the kidneys showing light and severe tubular hyaline degeneration (SDHD and white arrow), glomerular degeneration (GD), and increased capsule space of bowmam (IBC).

**Figure 6 pharmaceuticals-12-00173-f006:**
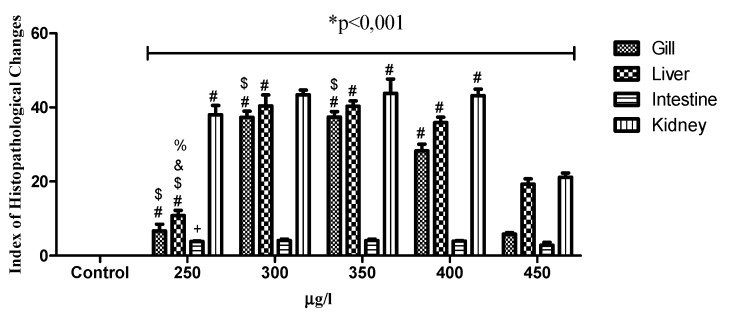
Index of histopathological changes for each concentration. Data are presented as mean ± SD (*n* = 4/groups); * *p* < 0.vs. control, # *p* < 0.001 vs. 450 µg/L, $ *p* < 0.001 vs. 400 µg/L, & *p* < 0.001 vs. 300 µg/L; % *p* < 0.01 vs. 450 µg/L, + *p* < 0.05 vs. 400 µg/L. Statistical analysis was performed through one-way ANOVA followed by the posthoc Tukey test.

**Table 1 pharmaceuticals-12-00173-t001:** Effect of the treatments with the different oral doses of EHFAo on behavioral reactions in zebrafish.

Group	Stage I	Stage II	Stage III	Total	%	Médian
Control	0/3	0/2	0/4	0/9	0	0.0 ± 0.0
44.457 mg/kg	1/3	2/2	3/4	6/9	66.6	58.6 ± 1.97
88. 914 mg/kg	1/3	2/2	3/4	6/9	66.6	59 ± 2.58
199.53 mg/kg	2/3	2/2	3/4	7/9	77.7	76.9 ± 2.22
218.83 mg/kg	2/3	2/2	4/4	8/9	88.8	87.6 ± 1.99
448.81 mg/kg	3/3	2/2	4/4	9/9	100	100 ± 0.0

**Table 2 pharmaceuticals-12-00173-t002:** Effect of the treatments with the different water concentrations of EHFAo on behavioral reactions in zebrafish.

Group	Stage I	Stage II	Stage III	Total	%	Median
Control	0/3	0/2	0/4	0/9	0	0.0 ± 0.0
250 µg/L	2/3	2/2	3/4	7/9	77.7	75.2 ± 1.97
300 µg/L	3/3	2/2	3/4	8/9	88.8	86.9 ± 2.01
350 µg/L	3/3	2/2	3/4	9/9	100	100 ± 0.0
400 µg/L	3/3	2/2	4/4	9/9	100	100 ± 0.0
450 µg/L	3/3	2/2	4/4	9/9	100	100 ± 0.0

**Table 3 pharmaceuticals-12-00173-t003:** Percentage of histopathological changes present in the tissues of animals treated orally with EHFAo.

Group/Tissue		Control	44.457 mg/kg	88.914 mg/kg	199.53 mg/kg	281.83 mg/kg	448.81 mg/kg
Gill	Total changes	0/22	3/22	4/22	4/22	4/22	3/22
	%	0	13.6	18.1	18.1	18.1	13.6
	Median	0 ± 0.0	0.66 ± 0.14	0.81 ± 0.14	0.75 ± 0.25	0.91 ± 0.14	0.66 ± 0.14
Liver	Total changes	0/20	10/20	12/20	15/20	17/20	15/20
	%	0	50	60	75	85	75
	Median	0 ± 0.0	29.9 ± 1.60	33.5 ± 1.37	39.3 ± 2.37	40.5 ± 3.10	12.7 ± 3.78
Intestine	Total changes	0/21	11/21	13/21	11/21	13/21	19/21
	%	0	52.3	61.9	52.3	61.9	90.4
	Median	0 ± 0.0	26.9 ± 0.62	29.2 ± 1.32	32.4 ± 1.37	32.7 ± 1.32	39.5 ± 1.56
Kidney	Total changes	0/22	11/22	13/22	14/22	17/22	13/22
	%	0	50	59	63.6	77.2	59
	Median	0 ± 0.0	12.9 ± 1.46	40.9 ± 2.87	38.4 ± 1.46	41.0 ± 3.78	13.3 ± 1.30

Data are presented as mean ± SD (*n* = 4/groups).

**Table 4 pharmaceuticals-12-00173-t004:** Percentage of histopathological changes present in the tissues of animals treated by immersion with EHFAo.

Group/Tissue		Control	250 µg/L	300 µg/L	350 µg/L	400 µg/L	450 µg/L
Gill	Total changes	0/22	17/22	21/22	19/22	11/22	9/22
	%	0	77.2	95.4	86.3	50	40.9
	Median	0 ± 0.0	6.6 ± 1.79	37.3 ± 1.66	37.4 ± 1.46	28.2 ± 1.80	5.8 ± 0.38
Liver	Total changes	0/20	8/20	16/20	15/20	14/20	18/20
	%	0	40	80	75	70	90
	Median	0 ± 0.0	10.8 ± 1.77	40.1 ± 2.96	40.3 ± 1.44	35.9 ± 1.46	19.3 ± 1.37
Intestine	Total changes	0/21	7/21	9/21	12/21	12/21	11/21
	%	0	33.3	42.8	57.1	57.1	52.3
	Median	0 ± 0.0	3.83 ± 1.44	4.08 ± 3.81	4.08 ± 3.81	3.91 ± 1.44	2.83 ± 0.76
Kidney	Total changes	0/22	22/22	17/22	19/22	17/22	13/22
	%	0	100	77.2	86.3	77.2	59
	Median	0 ± 0.0	38.0 ± 2.50	43.4 ± 1.25	43.8 ± 3.82	43.1 ± 1.77	21.1 ± 1.15

Data are presented as mean ± SD (*n* = 4/groups).

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
