# Peer review of "Acute Toxicity of the Hydroethanolic Extract of the Flowers of Acmella oleracea L. in Zebrafish (Danio rerio): Behavioral and Histopathological Studies"

_pharmaceuticals, 2019, doi:10.3390/ph12040173_

Round 1

Reviewer 1 Report

This paper reports on the zebrafish adult behavioral and histopathological changes induced by an hydroethanolic extract (HE) from the flowers of Acmella oleracea, a plant belonging to the Asteraceae family and widely used in Legal Amazon in local dishes and in folk medicine. Following exposure to a series of varying concentrations and using two administration methods, authors characterized behavioral phenotypes and evaluated histopathological lesions, concluding that the normal function of the organs was compromised in a dose and treatment-dependent manner.

Although this novel and original manuscript present interesting findings, as revision of the English is mandatory as some sentences are not clear. In addition, some doubts arose while reading the manuscript and changes are suggested:

Abstract:

1. The abstract should be reviewed to include a further description of the methodology used as well as the results obtained.

Introduction:

            1. Line 51 should be merged with paragraph of line 45.

            2. The objective of the work should be reviewed to include a brief description of the techniques used to reach the proposed aim.

Results:

1. Please include the chromatogram of the HE extract.

2. Review the order of tables and figures throughout the manuscript (e.g. line 77, tables should be numbered as 1 and 2). Also the order of the results should be reviewed. It seems that mortality was assessed before behavior and in the manuscript they appear in the opposite order.

3. Relative to the statistics, please include a multiple comparison post-hoc test in order to compare not only with the control group but also between concentrations. Include the p-values for the post-hoc test/comparisons done.

4. Review table 5 and 6. What is “stadio”? Is it the same as the stage described in the methods?

5. The manuscript presents too many figures and tables. In this regard, figure 3 and 5 as well as tables 7 and 8 could be moved to supplementary material maintaining only the figure with the histopathological index.

6. In figure 4, review the use of SEM. It seems that no error bar is present. The same in figure 6.

7. Review the use of SI units (e.g L instead of l).

Discussion:

1. The discussion should be reviewed. For instance, the first paragraph of the discussion should include a brief description of the objective of the work and a description of the most noticeable results obtained.

2. The second paragraph should also be reviews as an excessive number of references is used.

3. A more balanced discussion is required. In the current format, the authors present 16 paragraphs discussion histopathological changes and only 1 described and discussion behavioral changes. Moreover, nothing is discussed in relation to the LD50 and the LC50 calculated in this manuscript which are toxicological indicators of acute toxicity and important for establishing toxicological properties of this extract/plant. Besides that, instead of discussing each lesion in separate, it is suggested to interconnect all lesions and discuss them in a broad perspective and then progressively narrow down to the issue addressed in the paper.

Material and methods:

1. Please include a more detailed procedure for the chromatographic method.

2. Why was the yield so low? Usually, extraction of phenolic compounds from similar plant material results in yields around 20%. Is this the best extraction method? Is it already published? If so, please include the reference for the extraction method.

3. Please include the parameters related to the quality of the water (temp, pH, conductivity, etc).

4. How were the animals weighted? Also how were they immobilized in the wet sponge?

5. Rather than include the concentrations in topics, include them in the text in the SI units and refer that oral ones were prepared in saline and immersion in water from the maintenance system.

6. Review the sentences presented in this section as some make no sence. (e.g. “The animals were weighed, immobilized in a 7 cm wet sponge, prepared for this purpose, and thus, with the aid of a volumetric pipette (HTL Lab Solutions), the treatment with the doses of the EHFAo, at a maximum volume of 1.5 μl/animal [65,66].”

7. Include a more detailed description of the histopathological procedure.

8. Table 1, 2, 3 and 4 should be moved to supplementary material.

9. Relative to the statistics, review the use of SEM instead of SD and also check the normality of data before using parametric tests.

Conclusion:

1. Review the conclusion of the manuscript to not include results.

References:

1. Review the references according to the guide for authors. Some present the journal name in the abbreviated format, other present the name abbreviated with dots, etc.

Author Response

Attached are all comments for reviewers.

Reviewer 2 Report

Acmella oleracea is a kind of traditional medicine which improves broad spectrum of human diseases. The authors evaluated in vivo toxicity of hydroethanolic extract of Acmella oleracea flowers (EHFAo) using adult zebrafish by oral and exposure administration. Their histological analysis is quite detailed in each tissue and way of administration; however, their description is confusing and need to be improved. I recommend to use subheadings in the Results and the Discussion section.

Major comments:

The authors found a wide range of in vivo toxicity of Acmella oleracea flowers by using zebrafish. That’s fine, but in the introduction section, “the flowers are most used as a local anesthetic in toothaches (line 54)”. So, what is the relationship between the zebrafish results and human clinicals, especially anesthetic? Similar to the above, please compare the administration volume in zebrafish experiment and the amount of human intake in clinical. Otherwise, the reader cannot understand why such toxic extract can be used as a traditional medicine.

Minor comments:

Throughout the manuscript, some references seem not relevant so much for the story, while their previous “Acmella oleracea – zebrafish” paper is not in the references (de Souza GC, et al. Acmella oleracea (L) R. K. Jansen Reproductive Toxicity in Zebrafish: An In Vivo and In Silico Assessment. Evid Based Complement Alternat Med. 2019;1237301). And too many “Borges et al” in the text. Please re-arrange the reference list. Figure 1A (behavioral analysis of exposed EHFAo) should be Figure 1B, and Figure 1B (behavioral analysis of oral EHFAo) should be Figure 1A. I mean change order of subfigures. Figure 2, 4 and 6: Please add error bars in each graph. And in Figure 6, I cannot understand the indication of “p < 0.001” above the graph. Please insert scale bars in Figure 3 and 5, and unify the magnification of each image between oral administration and exposure, e.g., Figure 3B and 5B. Or are they the same size? (The same to “Figure 3C and 5C” and “Figure 3J and 5J”). Table 5 (behavioral analysis of oral EHFAo) should be renamed to Table 1, and Table 6 (behavioral analysis of exposed EHFAo) should be renamed Table 2. And re-order other Table numbers. Line 69-73: Please present the HPLC chromatograms. Line 121: the results were not statistically significant (p<0,001).

“p < 0.001” is quite significant. Please correct or reconsider this description.

Line 140: What is “MEV”? Line 140: In Figure 3J, white arrow indicates “dilatation of lamilae epitherium”, but I cannot understand the phenotypic change in this area. Normal images of each tissue would be a great help for readers to understand the toxic effect of EHFAo at glance. Line 341: Hyperemia is interesting. Are there any “vessel rapture” symptoms in the histological analysis in Figure 3 and 5? And ref 26 is not related with vessel rapture. Line 379-383: It’s hard to understand how much volume the authors administrated to zebrafish orally, and how to prepare working solution. Please update. And they use 4 animals x 5 groups x 3 replicates in the experiment, that means 12 animals per each experimental condition (n = 12). But the Table 5 and 6 showed n = 9. Why? The same problem exists in Table 7-8.

Author Response

Please see attachfile.

Round 2

Reviewer 1 Report

The authors have made several changes to the manuscript greatly improving its overall scientific quality. However, some changes were not made according to the previous review and thus the same comments are attached:

1- The presented chromatogram is the same as presented in another already published manuscript. Please include it in the supplementary material and change the text accordingly.

2 - The post-hoc statistics are still missing and results are only compared to the control group.

3 - Review tables and change "stadio" to stage.

4 - Review the use of SEM and change it to SD. SEM is not a descriptive statistics and should not be used as such.

5 - Statistical analysis is missing in the histopathological tables/results.

6 - Review the use of SI units (e.g L instead of l).

7 - Review line 244 of discussion. what is the relevance of this paragraph to the objective of the manuscript and what are the relation between the pharmacological properties of this major compound and the results obtained? Authors should comment on that and all the discussion/results obtained should be related to the chemical composition of the HE extract.

Author Response

The answer is attached. 

Round 3

Reviewer 1 Report

The authors have made substantial improvements to this manuscript by inclusion of new information and the comments suggested by the reviewer. Nonetheless some minor issues remain to be clarified:

Change the Figure 1 to Supplementary figure 1 and change the other accordingly. In figure 2 data is expressed as mean and SD? The info is missing. Also, include statistics and review the SI units used in the axis. Statistics in Table 1 and 2 are missing. Please include. In table 3, change commas to dots.
